# Phytochemical and Biological Characteristics of Mexican Chia Seed Oil

**DOI:** 10.3390/molecules23123219

**Published:** 2018-12-06

**Authors:** Yingbin Shen, Liyou Zheng, Jun Jin, Xiaojing Li, Junning Fu, Mingzhong Wang, Yifu Guan, Xun Song

**Affiliations:** 1Department of Food Science and Engineering, Jinan University, Guangzhou 510632, China; shenybin412@gmail.com (Y.S.); juningf0313@163.com (J.F.); 2State Key Laboratory of Food Science and Technology, Synergetic Innovation Center of Food Safety and Nutrition, School of Food Science and Technology, Jiangnan University, 1800 Lihu Road, Wuxi 214122, China; liyou890513@sina.com (L.Z.); zgzjjin@126.com (J.J.); Lixiaojing19900810@163.com (X.L.); 3Shenzhen Kivita Innovative Drug Discovery Institute, Shenzhen 518110, China; mzwang2000@163.com; 4Research Institute for Marine Drugs, Guangxi University of Chinese Medicine, Nanning 530200, China; 5School of Pharmaceutical Science, Shenzhen University, Shenzhen 518060, China

**Keywords:** Chia seed oil, polyunsaturated fatty acid, antioxidant, lipid-lowering effect

## Abstract

The purpose of this research was to investigate the chemical profile, nutritional quality, antioxidant and hypolipidemic effects of Mexican chia seed oil (CSO) in vitro. Chemical characterization of CSO indicated the content of α-linolenic acid (63.64% of total fatty acids) to be the highest, followed by linoleic acid (19.84%), and saturated fatty acid (less than 11%). Trilinolenin content (53.44% of total triacylglycerols (TAGs)) was found to be the highest among seven TAGs in CSO. The antioxidant capacity of CSO, evaluated with ABTS^•+^ and DPPH^•^ methods, showed mild antioxidant capacity when compared with Tocopherol and Catechin. In addition, CSO was found to lower triglyceride (TG) and low-density lipoprotein-cholesterol (LDL-C) levels by 25.8% and 72.9%respectively in a HepG2 lipid accumulation model. As CSO exhibits these chemical and biological characteristics, it is a potential resource of essential fatty acids for human use.

## 1. Introduction

Vegetable oils such as palm, canola, soybean, rapeseed, corn, olive, and sunflower oils are commonly used in cooking and industrial food manufacturing. The chemical constituents and content of cooking oils, particularly the type of fatty acids, are considered important criteria for quality of oils and their health benefits. Nutrition studies indicate that the type and content of dietary fat intake is closely related to various diseases, such as cardiovascular disease, diabetes, and depression [1]. Thus, consumption of healthier polyunsaturated fatty acids (PUFAs), which are considered ‘good fat’, is a vitally important prevention strategy for fat related health complications. However, as PUFAs cannot be synthesized in the human body, it is essential to supplement them through diet to meet daily needs [2]. Due to the huge market of PUFA rich oils, a number of industrial food manufacturing plants tend to seek alternative, plant-based sources of PUFAs.

Chia, botanically known as *Salvia hispanica* L., native to southern Mexico and northern Guatemala, is a 2 m tall herbaceous plant that belongs to the Lamiaceae family [3]. Currently, chia is grown on a commercial scale in the South America, Australia, Europe, and Southeast Asia, due to the high edible value in whole seeds, flour, and oil in food industry. In general, the major constituents in chia seeds include 25–40% oil, 17–24% protein, and 18–30% dietary fiber [4]. It also contains significant quantities of minerals and phytonutrients, including tocopherols, carotenoids and phytosterols, and phenolic compounds [5]. Chia seed oil is probably one of the healthiest oils on the market, containing predominantly essential fatty acids such as α-linolenic acid and linoleic acid. Numerous findings have confirmed that chia seed oil is a healthy oil for lowering the risk for cardiovascular disease, hepatoprotective effect, inflammation, and prevention of obesity-related disorders [6]. At present, the in vitro cancer cytotoxic properties of CSO have been reported by Ramzi et al., who showed that the CSO significantly inhibited the proliferation of human lymphoblastic leukemic cell lines, HeLa, and MCF-7 cells [7]. Up to now, there has been no evidence of adverse effects of chia seeds, and toxicological data on CSO from animal and controlled human trials on the safety and efficacy are still limited. However, experience gained from previous and current use of chia seeds for food purposes in developing countries can be regarded as supportive evidence to allow a positive conclusion on the safety of CSO [8].

Research findings by Ayerza et al. indicated that the content of the bioactive nutrients of chia seed can be affected by the geographical location and climate condition [9]. To date, research has been carried out on basic chemical and physical characteristics, including quantification of fatty acids, tocopherol, and polyphenols. However, to the best of our knowledge, sn-2 fatty acid composition and minor components of Mexican chia seed oil have not been extensively studied. The purpose of this research was to elaborate on most chemical compositions (fatty acids, triacylglycerol, tocopherols, sterols, polyphenols, metal elements, and PAHs) and functional values (antioxidant activity and lipid-lowering effects) of oil extracted from chia seeds grown in Mexico, to further evaluate its nutritional value and thus contribute towards identification of a potential food with medicinal and industrial applications.

## 2. Results and Discussion

### 2.1. Physical and Chemical Profiles

The physical and chemical properties analysis of CSO are presented in Table 1. The oil content of chia seed ranged from 31.39 to 32.39 g/100 g, much higher than that in soybean (17.6 to 25.4 g/100 g) reported by Dornbos et al. [10]. The result was consistent with the published data (31.2 g/100 g, on average, of Brazil chia seed) [11] and 28.5–32.70 g/100 g in chia seeds from Colombia, Argentina, Peru and Bolivia [12]. The results obtained in this study suggested that chia seed was a good source of crude oil based on the high yield of oil content.

### 2.2. Oxidative Stability

The induction period, widely used to determine stability of edible oils, is an indicator of oxidative processes. The oxidative stability of oils was measured as the induction period in response to forced oxidation. The results in Table 1 indicate that the induction period of CSO was found to be 0.68 h, which is similar to the reported value for Camelina oil (0.63 h) [13], but much lower than that of soybean oil [14]. The possible reason for lower induction period is the high content of unsaturated fatty acids in CSO (89.84% of total lipids). However, as there is limited data available on the stability of CSO, there is rising need for investigation on the stability of CSO during processing and storage.

At present there aren’t any color standards for CSO, and thus the Lovibond colorimeter was used firstly for color measurement. The color of CSO, represented by R and Y Lovibond scale, displayed more yellow (13 units) than red (1.65 units), which was consistent with that reported by Maira et al. [15]. The color ratio of CSO was also similar to that of linseed oil with yellow (70 units) and red (8.6 units) [16]. Nevertheless, CSO has different color than oils such as palm (yellow, 3.2 units; red, 27.4 units) and soy oil (yellow, 4.6 units; red, 10.6 units), which show more red than yellow [17]. Total pigment and carotenoid content are responsible for the natural color of vegetable oil, which could be an indicator of metamorphic oils. Significant changes of the R and Y values were found on the CSO during the metamorphic process, which could be used as a mark in distinguishing the degree of metamorphism.

### 2.3. Fat and Triacylglycerol Composition

As shown in Table 2, the most abundant fatty acid was α-linolenic acid (63.64% of total lipids), followed by linoleic acid (19.84% of total lipids), palmitic acid (7.07% of total lipids), and oleic acid (5.5% of total lipids). Among the fatty acids in CSO, the order of abundance of the identified components: α-linolenic acid > linoleic acid > palmitic acid > oleic acid > stearic acid > vaccenic acid > trianoic acid > arachidic acid > palmitoleic acid (Table 2 and Appendix A). This study showed that the major type of fatty acid in CSO was polyunsaturated fatty acid (PUFA, 89.84%). Interestingly, some research has reported that animal feed with chia seed of high PUFA content can increase the level of PUFA in meat fats, as well as aroma and flavor [18].

CSO has very high contents of α-linolenic (63.64%) and linoleic acids (19.84%), together accounting for more than 83%. In addition, the level of palmitic acid was 7.07%. We found that the ratio n-3/n-6 in CSO was 3.21, in agreement with the range of 3.18–4.18 reported by Ixtaina et al. [19], which is much higher than that of most vegetable oils, such as canola oil (0.45), olive oil (0.13), soybean oil (0.15), and walnut oil (0.20) [20]. Furthermore, the sn-2 fatty acid composition of CSO is indicated in Table 1 and Appendix A. Obviously, α-linolenic, linoleic, and oleic acids in CSO were the major sn-2 fatty acids, which were responsible for more than 95%. This result is in line with the recognized principle that unsaturated fatty acids mainly occupy the sn-2 position.

Based on a daily nutrient criterion for linoleic acid and α-linolenic acid proposed by National Institutes of Health [21], CSO could be applied as a good dietary supplement. In addition, CSO was characterized by a high polyunsaturated fatty acid/saturated fatty acid (PUFA/SFA) ratio, which is highly favorable for the reduction of serum cholesterol and atherosclerosis, and the prevention of cardiovascular disease [22]. Many studies have shown that CSO plays a role in lowering serum triglyceride, and also raises the level of HDL-C in rats [23]. As shown in Table 1, the PUFA/SFA ratio of CSO was 8.85. Thus, the incorporation of CSO into the diet could bring great beneficial effects to the cardiovascular system due to the high content of PUFAs.

The compositions of acylglycerol and free fatty acids are indicated in Table 2. The main composition of CSO was triacylglycerol, with its content up to 82.60%. The content of total diacylglycerol and FFA were 1.56% (1,3-DAG = 0.82% and 1,2 or 2,3-DAG = 0.74%) and 15.18%, respectively. As shown in Table 2, seven different TAGs were found in CSO, including aLnaLnaLn, aLnaLnL, aLnLL, aLnaLnP, aLnLO, aLnOP and aLnOO (Appendix A). The main triacylglycerols in CSO were aLnaLnaLn, aLnaLnL, aLnLL and aLnaLnP, and their levels were 53.44%, 23.76%, 8.22% and 6.25%, respectively. We found that α-linolenic acid was present in all the measured TAGs, which was also discovered by Ixtaina and coworkers [19].

### 2.4. Tocopherol and Phytosterol Levels

Tocopherols are natural antioxidants which can stabilize oils. The α-, γ- and δ-tocopherols, and total tocopherol content, in CSO are shown in Table 2 and Appendix A. CSO contained 76.96 mg/kg of tocopherols, mainly γ-tocopherol (>91%) and α-tocopherol (6.6%). δ-Tocopherol was present in low concentration (1.48 mg/kg). However, β-tocopherol was not detected in CSO. The content of total tocopherols (76.96 ± 8.47 mg/kg) in CSO was much lower than that of research (238–427 mg/kg) reported by Matthäus [24]. This difference may be explained by different varieties, processing, and storage conditions.

Table 2 and Figure 1A also show the content and the composition of sterols in CSO. Three phytosterols including campesterol, stigmasterol, and β-sitosterol were identified, with β-sitosterol as the most abundant sterol, accounting for more than 81% of the total amount of sterols. β-Sitosterol was predominant (2433.56 mg/kg), followed by campesterol (387.77 mg/kg) and stigmasterol (177.47 mg/kg). The total phytosterols of the CSO amounted to 3 g/kg, lower than the previous data ranged from 7 to 17 g/kg [25], which is probably linked to the loss of sterols during oil refining. Compared with other common edible oils, CSO contained a higher total tocopherol content than olive, soybean, peanut, corn, sunflower, and canola oils (260–1000 mg/kg) [26], indicating that CSO probably has relatively better oxidative stability.

It is well known that squalene is an important contributor to reduction of cholesterol levels. In this study, squalene was found in CSO at 226.43 mg/kg, much higher than the concentrations in walnuts, almonds, peanuts, hazelnuts, and macadamia nuts revealed by Maguire et al. [27]. The range of squalene content in CSO was 50–500 mg/kg, dependent on the extraction method used [27]. The high content of squalene in CSO is probably an important contribution to the beneficial effects of cancer prevention and health in human diet.

### 2.5. Determination of Trace Element Levels

Trace elements like manganese (Mn) increase the rate of oxidation of oil by the formation of free radicals of fatty acids and hydroperoxides [28], so they are undesirable in oils. The content of trace elements in CSO are shown in Table 2. Aluminum (Al), magnesium (Mg), calcium (Ca), boron (B), zinc (Zn), manganese (Mn), strontium (Sr) and arsenic (As) were detected in CSO. As indicated in Table 2, mineral composition of chia seed oil is low in elements, like all other oils (Ca = 1.226 mg/kg; Mg = 3.566 mg/kg; Zn = 0.153 mg/kg; Al = 4.104 mg/kg; Mn = 0.098 mg/kg; B = 0.193 mg/kg; Sr = 0.071 mg/kg). The most abundant trace element was aluminum (4.104 mg/kg). CSO is a rich source of minerals such as Ca, Mg and Zn. There were no toxic elements (Ni, Pb, Cd, Tl and Hg) detected except arsenic. However, mean levels of arsenic (0.014 mg/kg) do not raise concern. It is possible that concentrations of some elements were influenced by the growth, manufacturing process, and equipment. As arsenic was found in CSO, further investigation would be needed to determine its valence state, since different valence states vary in toxicity. All metals measured in CSO were lower than the maximum level accepted for virgin vegetable oils [29], and also lower than those in crude and degummed sunflower oils [30]. Thus, it could be used in food, medical and cosmetics industry.

### 2.6. Determination of Polycyclic Aromatic Hydrocarbons (PAHs)

PAHs constitute the critical hazards which are widespread in vegetable oil such as sunflower, olive, peanut, soybean, corn, canola, and palm oils. PAHs might occur from raw materials or the refining process [31]. Table 3 indicates the levels of the PAHs detected in CSO. Ten PAHs were revealed in CSO, including pyrene, fluoranthene, BaA, BbFlu, DBahA, Chr, BaP, BkFlu, IP and BghiP. Pyrene had the highest level (180.24 μg/100g) of the PAHs found in this oil. The concentrations of fluoranthene, BaA, and BbFlu were 84.72, 66.92, and 31.49 μg/100g, respectively. The total concentration of the other 6 PAHs was 45.18 μg/100g. The average sum of PAHs in CSO (41.06 μg/kg) was found to be higher than that in sunflower oil (17.36 μg/kg), and lower than that in soybean oil (65.33 μg/kg) [32]. One reason for PAHs introduced in vegetable oils is because of the heating-drying and extraction processes on oilseed or raw material [33]. Maximum concentration for total PAHs in edible oil in Regulation (European Union) is 10 μg/kg [34]. Thus, using uncontaminated raw materials, improving the refining process, and using activated carbon to remove PAHs during the oil refining process, are necessary requirements to decrease PAHs before market use [35].

### 2.7. FT-IR Spectrum Analysis

The FT-IR spectrum of CSO in Figure 1B shows an absorption band at about 3000 cm^−1^ corresponding with C–H stretching, and we observed the =C–H, asymmetric and symmetric methyl groups at 3010.26, 2923.62 and 2853.43 cm^−1^, respectively [36]. The frequencies at 1461.04 and 1742.79 cm^−1^ were assigned to bending vibration of lipid CH_2_ groups and the ester carbonyl stretching (C=O) fatty acids, respectively. According to the study by Guillen and Cabo, the distinct weak peak at 1654 cm^−1^ refers to C=C stretching absorptions of disubstituted *cis*-olefins, which could be used to evaluate the level of total lipids and identify total unsaturation in the oil [37].

Absorption bands associated with (C–C(=O)–O) and (OC–C) coupled bonds stretching are usually very strong at 1159.58 and 1099 cm^−1^. The peak at 1099 cm^−1^ is related to C–O–C stretch of triglyceride ester linkage. The bands at 719.9 cm^−1^ are due to a combined vibration of *cis*-disubstituted olefins having seven or more carbon atoms, and CH_2_ out-of-plane rocking [36].

The FT-IR spectrum clearly indicates that CSO contains C=C, representing the unsaturated fatty acids, and ester functional groups. The high level of α-linolenic acid (~64% in CSO), with three double bonds, led to a high degree of unsaturation in the spectrum. It was interesting to note that characteristic absorption of OH at 3470 cm^−1^ is not found on the FT-IR spectrum, which suggests that the moisture content could be neglected in CSO.

FTIR is one of the most practical, non-destructive and relatively cost-efficient techniques to evaluate relative levels of unsaturated fatty acid in vegetable oils by their characteristic peaks. The integration area of absorption peaks at 3010 cm^−1^ and 1743 cm^−1^ were representative of the content of PUFAs and total lipid respectively. Furthermore, the percentage of unsaturated fatty acid could be calculated by the area ratio of 3010/1743 cm^−1^.The percentage of unsaturated fatty acid found in this study indicates that CSO contains a high concentration of UFAs, which provided the unsaturation observed using individual feature of the peak at 3010 and 1654 cm^−1^ [36].

### 2.8. Antioxidant and Cytotoxicity

The antioxidant capability of CSO was evaluated by the inactivation of DPPH and ABTS assays. The radical inhibition decreased after treatment with CSO, as shown in Table 4. The IC_50_ value (IC_50_ value is the concentration of the sample required to inhibit 50% of radicals) of DPPH of the chia seed oil was 33.94 mg/mL, while the IC_25_ value (IC_25_ value is the concentration of the sample required to inhibit 25% of radicals) of ABTS was 28.51 mg/mL. Overall, catechin and tocopherol demonstrated superior scavenging activity than CSO. Both the ABTS and DPPH methods as investigated in our study gave similar results as the study by Xuan et al. [38] and by Scapin et al. for ethanol extract of Brazil chia seeds (3.84 mg/mL) [38], and by Scapin et al. for a Brazil chia seed ethanol extraction (3.84 mg/mL) obtained by solvent extraction [39]. Compared with sunflower, safflower, canola, and soybean oil, CSO presented the lowest DPPH and ABTS radical scavenging capability [38].

Further, the cytotoxic property of chia seed oil was assessed in LNcap and HepG 2 cells. Our data indicated the IC_25_ values (IC_25_ value is the concentration of the sample required to inhibit 25% of cell growth) of CSO were up to 580 and 889.68 μg/mL in LNcap and HepG 2 cell lines for 48 h, separately, which means cell toxic effects can be excluded by this extract method on this cell line (Table 4).

### 2.9. The Effect of Chia Seed Oil on the Hepg2 Lipid Accumulation Model

Hepatic steatosis can be induced in HepG2 cells by exposing cells to pathophysiological levels of oleic-acid (OA) to mimic the influx of excess lipo-toxicity of free fatty acids in hepatocytes [40]. To date there have been no reports published indicating anti-hepatic lipogenic effect in OA-induced HepG2 Cells. Thus, this study was designed to evaluate the anti-hepatic lipogenic effect of chia seed oil in the selected in vitro model. In the present study, upon establishing this successful hepatic steatosis model, intracellular TG, TC, HDL-C and LDL-C levels were significantly increased (Figure 2) after 0.2 mM OA stimulation. CSO (500 μg/mL) exhibited no cytotoxicity on HepG2 cells (Table 4). As showed in Figure 2, significant decreases of TG and LDL-C levels by 25.8% and 72.9% were seen in the CSO treatment group, separately, compared with OA-group, similar to the tocopherol group. Moreover, the group treated with CSO (500 μg/mL) demonstrated a higher reduction of LDL-C than the tocopherol treated group. We have demonstrated that chia seed oil does not reduce the TC secreted in HepG2 cells, but increases the levels of cellular HDL-C. These results could be supplementary in vitro evidence to the in vivo studies by Ayerza et al., which indicated that chia seed diets for 4 weeks decreased TG levels and increased HDL-C cholesterol content in rat serum [41]. In terms of lipid lowering efficacy, this study indicates that chia seed oil is more effective at reducing triglycerides than total cholesterol, similar to tocopherol. Thus, chia seed oil has great potential as nutraceutical in the food industry with its TG and LDL-C reducing effects, and increasing effect on the secretion of HDL-C.

## 3. Materials and Methods

### 3.1. Chemicals

Standards of tocopherols, phytosterols and squalene were purchased from Sigma Aldrich (St. Louis, MO, USA). Triacylglycerols were obtained from J&K Scientific (Beijing, China). Target standard compounds of polycyclic aromatic hydrocarbons (PAHs, dissolved in acetonitrile, 200 μg/mL), including benzo(*a*)anthracene (BaA), chrysene (Chr), benzo(*b*)fluoranthene BbFlu, benzo(*k*)fluoranthene (BkFlu), benzo(*a*)pyrene (BaP), dibenzo(*a.h*)anthracene (DBahA), benzo(*g.h.i*)perylene (BghiP), and indeno(1,2,3-*cd*)pyrene (IP), were purchased from Sigma-Aldrich (St. Louis, MO, USA). Chemical solvents were of HPLC grade. Commercial detection kits of triglyceride (TG), total cholesterol (TC), low-density lipoprotein-cholesterol (LDL-C) and high-density lipoprotein-cholesterol (HDL-C) were obtained from Nanjing Jiancheng Bioengineering Institute (Nanjing, China).

### 3.2. Extraction of Chia Seed Oil

Chia seeds (*S. hispanica* L.) were purchased from Guangzhou Yongheng Biotechnology Company Co., Ltd (Guangzhou, China). The chia seeds (100 g) were ground, and extracted with 0.5 L of *n*-hexane at 60 °C for 4 h in three repetitions. The obtained extracts were then combined and concentrated using a rotavapor apparatus. The resulting oil was stored at −20 °C for further analysis.

### 3.3. Oil Content of Chia Seeds

Determination of chia seed oil (CSO) content was adopted by the China National GB Standard for Oilseeds (GB/T 14488.1-2008) [42]. The oil content was reported as g/100 g seed.

### 3.4. Chemical Properties

#### 3.4.1. Oxidative Stability

The oxidative stability of CSO was evaluated by the widely used Rancimat analysis [43]. Briefly, 3 g oil was added to the reaction vessel in triplicate, followed by heating to 120 °C. The samples were then dried under airflow at the rate of 20 L/h. A receiver with 60 mL of distilled water was used for collecting the effluent air, containing volatile organic acids from CSO. The conductivity of the water as the process of oxidation proceeded, and the induction period in hours (h) was automatically measured by the machine.

#### 3.4.2. Color Determination

Color measurement was performed via official AOCS method (Cc 13b-45) [44]. In brief, CSO was loaded to a 25.4 mm cell, and the R/Y value was recorded following the instructions of Lovibond PFX 995 (Tintometer, Amesbury, UK).

#### 3.4.3. Fatty Acid Composition

Fatty acid composition of CSO was detected as methyl esters by the following method [45]. Briefly, 100 mg CSO was dissolved in 2 mL potassium hydroxide solution (KOH, 10% *w*/*v* in MeOH) and placed in water at 85 °C for 45 min. As the solution cooled, 5 mL of H_2_O and 5 mL of hexane were added and mixed thoroughly. Then mixed solution was subjected to liquid–liquid partitioning 3 times, and hexane fractions were combined. The hexane fractions with unsaponifiable substances were then washed with 10% alcohol until neutral pH was obtained, and finally dried by rotary evaporator. The hexane extract containing unsaponifiable substances was kept at −20 °C until further analysis for determination of sterols and squalene.

Saponified fatty acids were extracted from the aqueous layer (previously acidified with 2 M HCl) three times using hexane. The hexane layer containing saponified fatty acids was dried, and dissolved in 2 mL Boron trifluoride-methanol solution (14% methanol). The mixture reaction was then incubated at 60 °C for 45 min to obtain methylation of fatty acids. Hexane was used to extract fatty acid methyl esters (FAMEs) from the cooled mixture. Analysis of FAMEs was further carried out by gas chromatography (7820A, Agilent, Santa Clara, CA, USA), coupled with a FID and a Trace TR-FAME capillary column (i.d. 0.25 μm, 60 m × 0.25 mm, Thermo Fisher, Grand Island, NY, USA). The analysis condition was programmed as follows: the injector and FID temperatures were kept at 250 °C, and initial oven temperature was held at 80 °C for 3 min. It was then increased to 215 °C (15 °C/min), and finally up to 215 °C (20 min hold time). Nitrogen was used as carrier gas at 1 mL/min, with a split ratio of 1:20. Retention times (RT) of fatty acids were compared with those of authentic standards, and contents of individual components were expressed in relative percentages.

#### 3.4.4. Sn-2 Fatty Acid Composition

Analysis of sn-2 fatty acids of CSO was conducted by the assay used in reference [46]. Briefly, 10 mg CSO was added to a reaction solution containing 1 mL of 1 M Tris buffer (pH 8.0), 0.25 mL of bile salts (0.05%), 0.1 mL of calcium chloride (2.2%), and 10 mg of pancreatic lipase. The reaction mixture was kept incubated at 40 °C, with shaking for 3 min. The reaction was stopped by adding 1 mL of 6 M HCl. Separation of above reaction product was performed by preparative TLC on silica gel, with hexane/diethyl ether/acetic acid (1/1/0.02, *v*/*v*/*v*) as the developing solvent system. The band responding to the 2-monoacylglycerol (2-MAG) on the TLC plate was scraped into a glass tube and extracted 3 times with diethyl ether. The 2-MAG was dried and methylated to esters following the same saponification process as above. The resultant product was finally analyzed by the same analytical method.

#### 3.4.5. Triacylglycerol Composition

Analysis of triacylglycerol components of CSO was conducted by the official assay (AOCS Ce 5c-93), with HPLC-ELSD (Agilent 1200, Agilent Technologies, Santa Clara, CA, USA) equipped with a C18 column (150 mm × 4.6 mm, 5 μm, Varian, Palo Alto, CA, USA) [47]. CSO was prepared in hexane at 2 mg/mL and analyzed using the following conditions. Briefly, the mobile phase consisted of acetonitrile and isopropanol with a flow rate of 0.8 mL/min. The initial mobile phase was held at 60% acetonitrile for 40 min, changed linearly to 55% (40–80 min), returned to 60% (80–85 min) and maintained at 60% for 5 min (85–90 min). High sensitivity was obtained by using ELSD temperature set at 55 °C with a gain value of 8, and gas flow rate of 1.5 mL/min. Peaks were identified by comparing their retention times with standard reference compounds. Triacylglycerols were confirmed by comparison of RT with their standards. Relative content was reported as a percentage.

#### 3.4.6. FT-IR Spectrum Analysis

The infrared spectrum of CSO was measured by Fourier transform infrared spectroscopy (FTIR) with a Vector 70 model FT-IR instrument (Bruker, Billerica, MA, USA) in the infrared region of 4000−400 cm^−1^. This enabled investigation of structural information with a resolution of 4 cm^−1^. The assignment of functional groups corresponding to IR absorption bands was performed by comparison with distinctive bands of functional groups of the chemicals in edible oils.

### 3.5. Minor Components

#### 3.5.1. Tocopherol and Phytosterol Levels

Analysis of the composition of tocopherols was performed by HPLC using Waters 2475 multi λ fluorescence detector (λ_Ex_ 293 nm; λ_Em_ 325 nm) (Waters, Milford, MA, USA) based on reported methods [48]. Briefly, the column used in this analysis was a DIOL column (100 mm × 3 mm, 7 µm, Varian, Palo Alto, CA, USA). CSO was dissolved in hexane at 0.25 mg/mL and injected directly into the HPLC column. The optimized conditions for the run were as follows: in a linear gradient elution with the flow rate of 0.5 mL/min, and solvents hexane/tetrahydrofuran (A, 98/2, *v*/*v*) and isopropanol (B) with the following gradient timetable: 0–40 min, 100% A; 40–45 min, 100–95% A; 50–51 min, 95–100% A; 51–60 min, 100% A. The identification and quantification of α-, β-, γ- and δ-tocopherols in CSO was determined by comparison with their retention times and the area of the standards of tocopherols, and the unit was expressed as mg/kg of CSO.

The analysis of sterols and squalene was carried out by GC-MS equipped with a flame ionization detector (FID) and using a DB-5MS capillary column (60 m × 0.25 mm, 0.25 µm, Agilent, Santa Clara, CA, USA). Gas chromatograph conditions were as follows: helium was used as carrier gas at the flow rate of 1 mL/min, then the split ratio was 1:100. Oven temperature program: initial temperature 200 °C for 1 min, then 200 °C to 300 °C at 1 °C/min, and maintained at 300 °C for 18 min; detector temperature set at 290 °C, injector temperature 290 °C, injection volume, 1.0 µL. Sterol and squalene were determined and quantified by the internal standard (5α-cholestane), and the contents were expressed as mg/kg.

#### 3.5.2. Determination of Polycyclic Aromatic Hydrocarbons (PAH)

PAH levels were analyzed by LC/MS/MS. Briefly, CSO (0.5 g) was added a 10 ml Pyrex tube, and mixed with 5 µL of internal standard (2 mg/mL) and 3 mL of *n*-hexane by shaking. SupelMIP solid-phase extraction (SPE, Anpel, Shanghai, China) was applied to extract PAHs from the oil. The mixture contents were fully loaded to the SPE column, to allow adhesion of PAHs to the membrane. The column was then washed with hexane (5 mL), and PAHs were eluted with methylene dichloride (10 mL) at a low speed. Eluent was then dried and dissolved in acetonitrile (200 μL) for further experiments.

LC/MS/MS with APCI Source (Agilent, Santa Clara, CA, USA) was applied to detect PAHs in CSO. Separation was achieved using a C18 column (100 × 2.1 mm, 1.8 µm; Agilent, Santa Clara, CA, USA) with a mobile phase comprising of solvent A (0.1% formic acid in acetonitrile) and solvent B (0.1% formic acid in water). The column oven temperature was set at 25 °C, and flow rate was 0.5 mL/min. The optimized conditions for separation were as follow: linear gradient elution was started from 45% A and then increased from 45 to 100% A (1–15 min), held 100% A from 15 to 21 min, next decreased 100% to 45% A from 21–22 min, lastly maintaining 45% A for 3 min. The PAH concentrations in CSO were quantified with internal standard.

#### 3.5.3. Determination of Trace Element Levels

The content of trace elements of CSO was determined by inductively coupled plasma-mass spectrometry (ICP-MS), based on previous methods with minor modifications [49]. Briefly, the oven temperature was programmed from 60 °C (3 min hold) to 175 °C at 5 °C/min, maintained at 175 °C for 15 min, then raised to 220 °C at 2 °C/min, and finally held at 220 °C for 10 min.

#### 3.5.4. Antioxidant Activity In Vitro

The in vitro antioxidant effects of CSO were evaluated by DPPH and ABTS assays [50]. Briefly, 50 μL volumes of different concentrations of CSO were added to 150 μL of DPPH solution (200 μM) with shaking. After incubation of 40 min, optical density was recorded at 517 nm (OD_517_). Tocopherol was used as a positive control. In the same manner, ABTS radical scavenging capability of CSO was measured at a wavelength of 734 nm. Catechin was used as the positive control in ABTS assay. The formula for calculating % scavenging effects (SE) by DPPH and ABTS was as follows:$$\mathrm{SE}\left(\%\right)=\frac{\mathrm{ODcontrol}-\mathrm{ODsample}}{\mathrm{ODcontrol}}\times 100\%$$

### 3.6. Cell Cytotoxicity

The cytotoxic effects of the chia seed oil against LNcap cells and HepG2 were tested using a 3-(4,5-dimethyl-2-yl)-2, 5 diphenyltetrazolium bromide (MTT) assay [51]. Cells were obtained from Stem Cell Bank, Chinese Academy of Sciences, Shanghai, China. Cells were cultured in Dulbecco’s modified Eagle’s medium (DMEM; Thermo, Grand Island, NY, USA) containing 10% fetal bovine serum (FBS; Gibco, USA) in an incubator (37 °C, 5% CO2). The dried CSO was dissolved in DMSO to 20 mg/mL and sterilized by filtration through a 0.22 μm membrane filter (Millipore, Burlington, MA, USA) as a stock solution. To each well of a 96-microwell plate, 190 μL of LNcap cells or HepG2 (5 × 10^4^ cell/mL) was added with or without 10 μL of different concentrations of CSO. After 48 h incubation, 20 μL of MTT (5 mg/mL) was added and then incubated for an additional 4 h. The supernatant was then removed completely, and 150 μL of DMSO was added to each well. Lastly, the OD value at 570 nm was measured using a microplate reader (Bio Tek, San Diego, CA, USA). Cell viability was measured and calculated by the formula: cell viability (%) = [OD _570_ (sample)/OD _570_ (control)] × 100%.

### 3.7. CSO Inhibits OA-Induced Intracellular Lipid Level in HepG2 Cells

#### 3.7.1. Cell Treatment

HepG2 was cultured in DMEM with high glucose (Gibco, USA) supplement with 10% FBS (Gibco, USA) at 37 °C with 5% CO_2_ in a cell incubator. The oleic acid (OA) was dissolved in 10% bovine serum albumin (BSA, fatty acid free, Sigma, St. Louis, MO, USA), and diluted in a culture medium with 1% BSA as the final concentration of 200 μM. CSO was prepared in DMEM as stock solution. The cells were seeded in 6-well plates and incubated for 24 h, before being cultured as follows: DMEM plus 1% BSA as a control group; DMEM with OA-BSA at 200 μM as a model group; CSO at final concentration of 500 μg/mL was applied with OA-BSA simultaneously as a co-treatment.

#### 3.7.2. Measurement of Cellular Lipid Levels in HepG2 Cells

Contents of TG, TC, LDL-C and HDL-C in HepG2 cell model were measured by commercial kits (Nanjing Jiancheng Bioengineering Institute, Nanjing, China). Briefly, cells were collected and washed twice with pre-chilled PBS to remove culture medium after treatment for 24 h. Then, 200 μL of 2% Triton X-100 was added for lysing cells by sonication. The supernatant was collected for analysis of TG, TC, LDL-C and HDL-C. Results were normalized to protein concentration using the Bradford protein assay kit (Bio-Rad, Hercules, CA, USA).

### 3.8. Statistical Analysis

Statistical analysis was carried out using GraphPad Prism 7.0 (GraphPad, San Diego, CA, USA). Differences were considered to be statistically significant at *p* < 0.05, or very significant at *p* < 0.01, by the Duncan test.

## 4. Conclusions

In conclusion, PUFA analysis in chia seed oil from Mexico was greater than 80% of fatty acids. This research indicates that α-linolenic acid was the most abundant fatty acid, followed by linoleic acid with an n-3/n-6 ratio of 3.21:1. Due to the large amount of easily-oxidized PUFAs in chia seed oil (induction period = 0.68 h), raw chia seeds might need special treatment during sample collection, processing, storage, and transportation, to prevent oxidation and maintain the quality of the oil. We also found that CSO presented some antioxidant effects in vitro, which are beneficial for human health. This is the first study to report the ability of CSO to reduce the levels of hepatic TG and LDL-C markedly, and increase the content of HDL-C in HepG2 cell model. Thus, this indicates that chia seed oil may help in reducing the risk for cardiovascular disease, and it can be widely used as cooking oil and in manufacture of healthy food supplements.

## Figures and Tables

**Figure 1 molecules-23-03219-f001:**
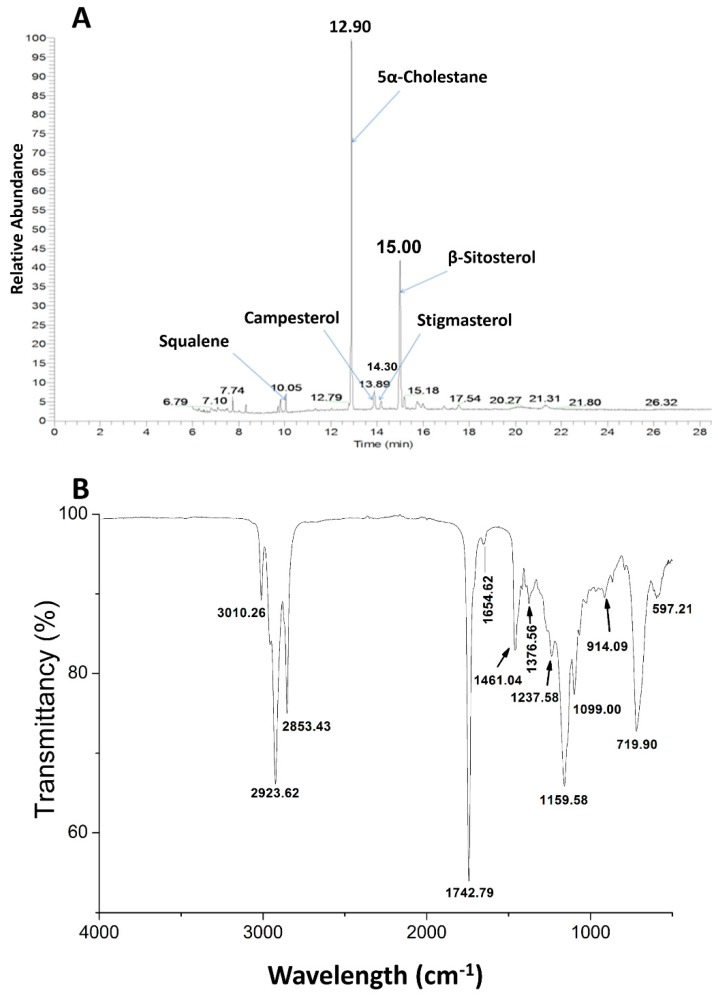
(**A**) Sterols in Mexican chia seed oil were analyzed using a gas chromatograph-mass spectrometer system by standards. (tR = 10.05 min), squalene; (tR = 12.90 min ), 5-α-cholestan; (tR = 13.89 min), campesterol; (tR = 14.30 min ), stigmasterol; (tR = 15.00 min), β-sitosterol. (**B**) FTIR spectra for chia seed oil extracted.

**Figure 2 molecules-23-03219-f002:**
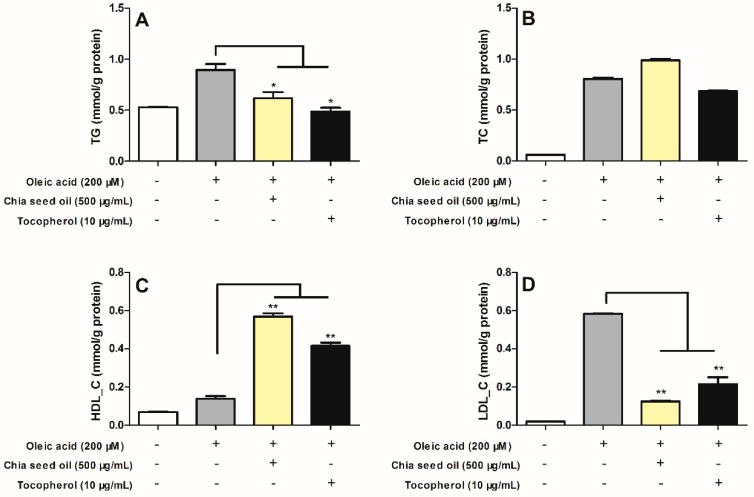
Effects of chia seed oil on accumulation of the intracellular (**A**) TG, (**B**) TC, (**C**) HDL-C, and (**D**) LDL-C in oleic acid (OA)-induced HepG2 cells. OA-induced accumulation of TG and LDL-C was significantly reduced by chia seed oil co-treatment in HepG2 cells. Vertical bars represent the mean ± SE of three independent experiments. * *p* < 0.05, ** *p* < 0.01 versus OA group.

**Table 1 molecules-23-03219-t001:** Physical and chemical characters of chia seed oil.

Oil Parameters	Content	
Oil content (g/100 g)	31.89 ± 0.50	
Oil stability (induction period)/h	0.68 ± 0.03	
Oil color/(units)		
	R	Y
	1.65 ± 0.07	13.00 ± 0.00
**^a^ Fatty Acid Composition**		
Palmitic acid C16:0	7.07 ± 0.01	
Palmitoleic acid C16:1 (n-9)	0.06 ± 0.00	
Trianoic acid C17:0	0.16 ± 0.01	
Stearic acid C18:0	2.81 ± 0.04	
Oleic acid C18:1 (n-9)	5.50 ± 0.01	
Vaccenic acid C18:1 (n-7)	0.80 ± 0.01	
Linoleic acid C18:2 (n-6)	19.84 ± 0.01	
α-Linolenic C18:3 (n-3)	63.64 ± 0.06	
Arachidic acid C20:0	0.12 ± 0.01	
SFA	10.16 ± 0.06	
PUFA	89.84 ± 0.07	
PUFA/SFA	8.85 ± 0.06	
n-3/n-6 FA ratio	3.21 ± 0.00	
**sn-2 Fatty Acid Composition**		
C16:0	1.10 ± 0.05	
C18:0	0.88 ± 0.09	
C18:1	6.38 ± 0.12	
C18:2	25.07 ± 0.06	
C18:3	63.76 ± 0.66	

^a^ Values reported as means ± SD of three replicate analyses (n = 3). SFA = total saturated fatty acids, PUFA = total polyunsaturated fatty acids, n-6 = total omega-6, n-3 = total omega-3 fatty acids.

**Table 2 molecules-23-03219-t002:** Fat, TAGs and minor components of chia seed oil.

Oil Parameters	Percentages
**Fat Compositions/%**	
TAG	82.60 ± 0.15
1,3-DAG	0.82 ± 0.03
1,2(2,3)-DAG	0.74 ± 0.02
Total DAG	1.56 ± 0.02
FFA	15.18 ± 0.11
**TAG Compositions/%**	
aLnaLnaLn	53.44 ± 0.47
aLnaLnL	23.76 ± 0.22
aLnLL	8.22 ± 0.24
aLnaLnP	6.25 ± 0.05
aLnLO	1.80 ± 0.24
aLnOP	4.43 ± 0.28
aLnOO	2.10 ± 0.35
Di-UTAG	10.69 ± 0.33
Tri-UTAG	89.31 ± 0.34
Minor components	^a^ Content (mg/kg)
**Tocopherols**	
α-tocopherol	5.10 ± 0.42
γ-tocopherol	70.38 ± 7.99
δ-tocopherol	1.48 ± 0.06
Total tocopherols	76.96 ± 8.47
Squalene	226.43 ± 38.19
**Phytosterols**	
Campesterol	387.77 ± 59.05
Stigmasterol	177.47 ± 31.57
β-Sitosterol	2433.56 ± 71.69
Total phytosterols	2998.80 ± 162.30
**Mineral contents**	
Boron	0.193 ± 0.012
Magnesium	3.566 ± 0.185
Aluminum	4.104 ± 0.644
Calcium	1.226 ± 0.082
Manganese	0.098 ± 0.010
Zinc	0.153 ± 0.017
Arsenic	0.014 ± 0.004
Strontium	0.071 ± 0.014

Tri-UTAG, triunsaturated triacylglycerols; Di-UTAG, diunsaturated triacylglycerols; P, palmitic acid; S, stearic; O, oleic; L, linoleic acid; Ln, linolenic acid; aLn, α- linolenic acid. ^a^ All the measurements are in terms of mean ± standard deviation.

**Table 3 molecules-23-03219-t003:** Polycyclic aromatic hydrocarbons (PAH) in chia seed oil.

PAHs	Concentration (μg/100 g)
Fluoranthene	84.72 ± 9.85
Pyrene	180.24 ± 18.84
Benzo (*a*) anthracene (BaA)	66.92 ± 10.05
Chrysene (Chr)	5.69 ± 0.98
Benzo (*b*) fluoranthene (BbFlu)	31.49 ± 8.84
Benzo (*k*) fluoranthene (BkFlu)	2.21 ± 0.95
Benzo (*a*) pyrene(BaP)	3.68 ± 1.02
Dibenzo (*a.h*) anthracene (DBahA)	30.04 ± 6.62
Indeno(1,2,3-*cd*) pyrene and Benzo (*g,h,i*) perylene (IP and BghiP)	3.56 ± 0.88

**Table 4 molecules-23-03219-t004:** Antioxidant activity and cytotoxicity of chia seed oil.

Samples	Antioxidant Activity		Cytotoxicity ^c^ IC_25_ (μg/mL)	
	DPPH ^a^ IC_50_ (mg/mL)	ABTS ^b^ IC_25_ (mg/mL)	LNcap	HepG2
Chia seed oil	33.94	28.51	580.12	889.68
Catechin	0.005	-	-	-
Tocopherol	-	0.004	-	-

^a^ IC_50_ value is the concentration of the sample required to inhibit 50% of radical of DPPH, ^b^ IC_25_ value is the concentration of the sample required to inhibit 25% of radical of ABTS, ^c^ IC_25_ value is the concentration of the sample required to inhibit 25% of cell growth. IC_25_ value was much more reliable in ABTS and cytotoxicity assay because it was in the range of tested concentration.

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
