# Peer review of "Phytochemical and Biological Characteristics of Mexican Chia Seed Oil"

_molecules, 2018, doi:10.3390/molecules23123219_

Round 1

Reviewer 1 Report

However, the paper presents good results, I have major concerns and comments that should be addressed to make the manuscript publishable

Introduction

The introduction section needs more improvements. For example, information regarding the potential cytotoxicity and the safety profile of CSO is missed. This point should be addressed and clarified with adequate references.

Line 48: ''Phytosterols'' the word is repeated in the statement. Please, delete it. 

Materials and methods

Important point is that the authors should provide in all experimental methods the solvent that has been used to dissolve the extracted CSO.

I can see that the authors did not provide the reference for the following used methods. However, there are well-known methods, but it is important to provide their references.

Determination of chia seed oil (CSO) content. Please, provide the reference of the used method.

Oxidative stability. Please, provide the reference of the used method.

Color determination. Please, provide the reference of the used method.

The fatty acid composition of CSO. The authors reported that fatty acid composition of CSO was detected as methyl esters. Did the authors develop this method? If not why did not the authors provide the reference of the used methods? Please clarify this point.

Again where is the reference for the methods used for the determination of triacylglycerol Composition?. Also, Based on what reason the authors have chosen standards of triacylglycerol for the analyses?

Again where are the references for the methods used for the determination of antioxidant activity and cell cytotoxicity?

 Please provide all chromatograms for all detected/identified compounds in CSO in a supplementary file, especially those obtained by HPLC and  GC/MS analyses. The authors provided only in Figure 1 the chromatographic data of sterols by GC-MS and FTIR spectra for extracted CSO. 

Results and discussion

Discussion of the results needs more improvement, where the results are poorly discussed. I recommend the authors to carefully revise this section with an emphasis on more adequate references to support their statements and rationalize the results with previously reported studies.

Table 4. Why did the author determine IC25 for evaluating the cytotoxicity of CSO instead of IC50? please clarify this point in the results and discussion section. 

Conclusions

Line 441. It would be better to state that ‘’Further studies are needed to evaluate the possibility of using CSO as cooking oil’’ instead of stating ‘’ it could be widely used as cooking oil…’’

In general, I recommend the authors to revise carefully their paper and to take into considerations all comments that I have addressed.

Author Response

Response to Reviewer 1 Comments

Point 1: The introduction section needs more improvements. For example, information regarding the potential cytotoxicity and the safety profile of CSO is missed. This point should be addressed and clarified with adequate references. 

Response 1: Thanks for the valuable suggestions. The information regarding the potential cytotoxicity and safety profile of CSO was added in the introduction, which was also highlighted in yellow.

Point 2: Line 48: ''Phytosterols'' the word is repeated in the statement. Please, delete it.

Response 2: The word ''Phytosterols'' was deleted.

Point 3: Materials and methods

Important point is that the authors should provide in all experimental methods the solvent that has been used to dissolve the extracted CSO.

Response 3: The solvent that has been used to dissolve the extracted CSO was provided detailly in the experimental methods.

Point 4: I can see that the authors did not provide the reference for the following used methods. However, there are well-known methods, but it is important to provide their references. Determination of chia seed oil (CSO) content. Please, provide the reference of the used method.

Response 4: References were added as suggested.

Point 5: Oxidative stability. Please, provide the reference of the used method.

Response 5: References were added as suggested.

Point 6: Color determination. Please, provide the reference of the used method.

Response 6: The reference was added as suggested.

Point 7: The fatty acid composition of CSO. The authors reported that fatty acid composition of CSO was detected as methyl esters. Did the authors develop this method? If not why did not the authors provide the reference of the used methods? Please clarify this point.

Response 7: Thanks for the suggestions. The method used was according to a literature, and the reference has been added as suggested.

Point 8: Again where is the reference for the methods used for the determination of triacylglycerol Composition? Also, Based on what reason the authors have chosen standards of triacylglycerol for the analyses?

Response 8: Thanks for the suggestions. Analysis of triacylglycerol components of CSO was performed by the official assay (AOCS Ce 5c-93), and the reference has been added.

We chose standards of triacylglycerol for the analyses based on the recommendation of this official assay.

Point 9: Again where are the references for the methods used for the determination of antioxidant activity and cell cytotoxicity?

Response 9: The references for the determination of antioxidant activity and cell cytotoxicity have been added.

Point 10: Please provide all chromatograms for all detected/identified compounds in CSO in a supplementary file, especially those obtained by HPLC and GC/MS analyses. The authors provided only in Figure 1 the chromatographic data of sterols by GC-MS and FTIR spectra for extracted CSO.

Response 10:Thanks for the suggestions. Some important chromatograms of fatty acids, sn-2 fatty acids, tocopherols and triacylglycerols detected/identified /obtained by HPLC are provided in the supplementary file (Figure S1-Figure S4).

Point 11: Results and discussion

Discussion of the results needs more improvement, where the results are poorly discussed. I recommend the authors to carefully revise this section with an emphasis on more adequate references to support their statements and rationalize the results with previously reported studies.

Response 11:The results are discussed further and more adequate references have been added in this section to support our statements.

Point 12: Table 4. Why did the author determine IC25 for evaluating the cytotoxicity of CSO instead of IC50? please clarify this point in the results and discussion section.

Response 12:Thanks for the valuable suggestions. Since CSO showed low cytotoxicity against LNcap and HepG 2 cell lines. The value of IC25 was in the range of the tested concentration, but IC50 value of CSO was out of the range of tested concentration. The software could only give IC50 predictions, and the IC25 is much more reliable than IC50. We have clarified this point in the table notes.

Point 13: Line 441. It would be better to state that ‘’Further studies are needed to evaluate the possibility of using CSO as cooking oil’’ instead of stating ‘’ it could be widely used as cooking oil…’’

Response 13:The sentence has been revised as suggested.

Others

All changes in the manuscript, figures and tables were highlighted in yellow.

Reviewer 2 Report

The document describes the chemical characterization and functional analysis of chia edible oil. The information is of importance to the area of chemical characterization of oils, as well as to propose mechanisms of health promotion with bioactive components.

However, there are some questions that need to be address

For one, how do the authors are confident that the chia analyzed was cultivated in Mexico? According to the document, they acquired the seeds form a company in China.

They also claim that there are only few documents on the composition of Chia oil, but there are several references found, including from other regions such as Argentina. This is important because the authors want to explain the differences in chia grown in different locations. Below, few references found.

Ayerza, R. (1995). Oil content and fatty acid composition of chia (Salvia hispanica L.) from five northwestern locations in Argentina. Journal of the American Oil Chemists’ Society72, 1079-1081.

Coates, W. (2011). Protein content, oil content and fatty acid profiles as potential criteria to determine the origin of commercially grown chia (Salvia hispanica L.). Industrial Crops and Products34(2), 1366-1371.

da Silva Marineli, R., Moraes, É. A., Lenquiste, S. A., Godoy, A. T., Eberlin, M. N., & Maróstica Jr, M. R. (2014). Chemical characterization and antioxidant potential of Chilean chia seeds and oil (Salvia hispanica L.). LWT-Food Science and Technology59(2), 1304-1310.

Discussion needs to be improved, since there are no comparison with other reports on the composition of chia oil.

References need to be carefully revised. .Reference 3 and 7 are the same one.

Please, revise the document for English

Author Response

Response to Reviewer 2 Comments

Point 1: For one, how do the authors are confident that the chia analyzed was cultivated in Mexico? According to the document, they acquired the seeds form a company in China.

Response 1: Thanks for the comments. There is no chia cultivated in China. The chia sample analyzed in our experiments was bought from an import company (Guangzhou Yongheng Biotechnology Company Co., Ltd, Guangzhou, China), which was imported from Mexico with official imported certificate.

Point 2: They also claim that there are only few documents on the composition of Chia oil, but there are several references found, including from other regions such as Argentina. This is important because the authors want to explain the differences in chia grown in different locations. Below, few references found.

Ayerza, R. (1995). Oil content and fatty acid composition of chia (Salvia hispanica L.) from five northwestern locations in Argentina. Journal of the American Oil Chemists’ Society, 72, 1079-1081.

Coates, W. (2011). Protein content, oil content and fatty acid profiles as potential criteria to determine the origin of commercially grown chia (Salvia hispanica L.). Industrial Crops and Products, 34(2), 1366-1371. da Silva Marineli, R., Moraes, É. A., Lenquiste, S. A., Godoy, A. T., Eberlin, M. N., & Maróstica Jr, M. R. (2014).

Chemical characterization and antioxidant potential of Chilean chia seeds and oil (Salvia hispanica L.). LWT-Food Science and Technology, 59(2), 1304-1310.

Response 2: Thanks for the valuable suggestions. There are some differences between the literatures listed and our research on CSO.  

In the study by Ayerza et.al, they just reported the oil content and fatty acids from the chia seed oil in in five Northwestern Argentina locations.

The research by Coates et. al reported the fatty acid composition, protein and oil content of chia seeds grown in some of the larger commercial fields in South America to provide how these properties relate to the characteristics of the ecosystems in which they were grown.

The research by da Silva Marineli, R. et al. was to chemically and nutritionally characterize the commercial chia seeds and oil from Chile and investigate their antioxidant potential by different in vitro methods.

Most literatures only reported the either chemical constitute of chia seed oil or its function. However, they did not report the trace elements and PAHs in the oil. While, the purpose of our work was focus on the chemical profile, nutritional quality, antioxidant and hypolipidemic effects of Mexican chia seed oil as food systematically. The study on the trace inorganic element levels in CSO and the effect of CSO on the HepG2 lipid accumulation has not been reported before. In summary, we systematically reported the chemical profile, nutritional quality, antioxidant and hypolipidemic effects of Mexican chia seed oil.

Point 3: Discussion needs to be improved, since there are no comparison with other reports on the composition of chia oil.

Response 3: Thanks for the valuable suggestions. The discussion has been improved, which is highlighted in the results and discussion, and some comparison with other reports on the composition of chia oil has been added in this section.

Point 4: References need to be carefully revised. Reference 3 and 7 are the same one.

Response 4: The references have been revised carefully.

Point 5: Please, revise the document for English.

Response 5: The document for English has been improved by English native speakers. All the writing improvement was in red.

 Others

All changes in the manuscript, figures and tables were highlighted in yellow.

Round 2

Reviewer 1 Report

The manuscript has been significantly improved. 

Reviewer 2 Report

suggestions incorporated